# Study on the Cooling Performance of a Focused Ultrasonic Radiator for Electrical Heating Elements

**DOI:** 10.3390/mi15010116

**Published:** 2024-01-10

**Authors:** Songfei Su, Yang Wang, Lukai Zheng, Mengxin Sun, Qiang Tang, Huiyu Huang

**Affiliations:** 1School of Mechanical Engineering, Nanjing Institute of Technology, Nanjing 211167, China; wangyang628729@163.com (Y.W.);; 2Jiangsu Key Laboratory of Advanced Manufacturing Technology, Faculty of Mechanical and Material Engineering, Huaiyin Institute of Technology, Huaian 223003, China

**Keywords:** focused ultrasound, acoustic streaming, heat dissipation, electronic device

## Abstract

In this work, a focused ultrasonic radiator is proposed for cooling the electrical heating elements in the focal region, and its working characteristics are investigated. The analyses of the FEM computational and flow field visualization test results indicate that focused ultrasound can generate forced convective heat transfer by the acoustic streaming in the focal region, which can cool the heating elements effectively. Experiments show that when the input voltage is 30Vp-p and the ambient temperature is 25 °C, the focused ultrasonic radiator can cause the surface temperature of the heating element (high-temperature alumina ceramic heating plate with a diameter of 5 mm) in the focal region to drop from 100 °C to about 55 °C. When the diameter of the electrical heating element is changed from 5 mm to 30 mm, the cooling effect is similar in the focal region. Compared with a fan, the focused ultrasound radiator has a shorter cooling time and a more concentrated cooling area. The focused ultrasonic radiator proposed in this work is suitable for some special environments.

## 1. Introduction

With the development of electronic technology, electronic devices are constantly miniaturized and integrated, and their power density is significantly grown. As a result, the heat generated per unit volume is increasing. In order to ensure the safe working circumstances for electronic equipment, the temperature rise of key devices or hotspots needs to be well controlled. Common methods for cooling electronic devices include heat sinks, liquid cooling system, axial fans, and so on [1,2,3,4]. However, the high integration, miniaturization, and application situation diversity of electronic devices put forward higher requirements (such as small area heat dissipation, high penetrating capability, or low noise) for cooling system designs [5,6,7]. An effective cooling method for small heating elements can not only reduce the structural complexity of cooling systems, but also make cooling systems more energy-efficient.

Ultrasound has several unique features in terms of ease of operation, flexible control, strong penetrability, and good biocompatibility. It has been widely used in material processing, medical diagnosis, weapons, defect detection, and other fields [8,9,10,11,12]. In recent years, ultrasound used for cooling have been proposed by many researchers [13,14]. The mainly physical effect of ultrasound used for cooling in air concerns acoustic streaming fields [15,16,17]. Numerical simulation analysis and experiments have verified that acoustic streaming fields generated by standing wave, traveling wave, or near field can enhance convection heat transfer [18,19,20,21,22]. In [21], a device for generating acoustic streaming from different ultrasonic vibration modes was developed to enhance convective heat transfer. When the vibration frequency was 30 kHz (the resonant frequency of the Langevin transducer) and the heat source was placed above the radiation surface, the plane longitudinal acoustic vibration could quickly cause acoustic streaming and lead to a significant decrease in temperature. The cooling performance was closely related to the distance between the radiation surface and the heat source, and the cooling efficiency was highest when the distance was an integer multiple of the ultrasonic half wavelength. In [22], the acoustic streaming generated by ultrasonic flexural vibration at a frequency of 28.4 kHz (the resonant frequency of the Langevin transducer) was employed to investigate the heat dissipation characteristics. By virtue of the acoustic streaming, a notable temperature drop of 40 °C was obtained in 4 min and maintained. Although ultrasound has an effective cooling effect, it is difficult to generate strong acoustic streaming.

The focused ultrasonic transducer can direct sound energy towards a specific location, resulting in a higher acoustic streaming velocity compared to other types of ultrasonic fields due to the increased Reynolds stress gradient [23,24]. Traditionally, focused ultrasound has been employed for the ablation of solid tumors at its focal area [25,26]. This study introduces a novel focused ultrasound-based cooling method that can effectively cool a heating element situated in the focal region through acoustic streaming generated using a focused ultrasonic radiator. Through FEM computation, a flow field visualization test, and experiments, the cooling principle of the focused ultrasonic radiator is analyzed, and the cooling effects are investigated and clarified. The cooling efficacy is compared to that of a commercial centrifugal fan. Utilizing focused ultrasound instead of other ultrasonic fields can significantly enhance acoustic energy utilization and improve the ultrasonic cooling effect.

## 2. Materials and Methods

Figure 1 shows the experimental setup to investigate the cooling effect and characteristics of the focused ultrasonic radiator proposed in this work. The experimental system is mainly composed of a focused ultrasonic radiator, a heating element, and two positioning platforms. The focused ultrasonic radiator is driven by an electrical driving system, which consists of a signal generator, an oscilloscope, and a power amplifier. The heating element is a circular ceramic heating plate with a diameter of 5 mm and a thickness of 0.5 mm. The temperature of the heating element is controlled by a DC power supply and measured by a thermometer (TA612C, K-type, Teansi Science and Technology Co., Ltd., Suzhou, China), which is in contact with the heating element surface and can measure the temperature at four points simultaneously. In the experiments, the heating element is located below the radiation face of the focused ultrasonic radiator, which is fixed onto a platform of a XYZ stage (LD60-LM-2, Jutie Precise Machinery Co., Ltd., Shenzhen, China) for adjusting the relative location of the heating element and focused ultrasonic radiator.

Figure 2 shows the structure and dimensions of the focused ultrasonic radiator, which consists of a Langevin transducer (HNC-4AH-2560, Hainertec Co., Ltd., Suzhou, China) with a first-order resonance frequency of 62.1 kHz, and a radiation unit with a concave radiation face. The radiation unit is made of aluminum and its apparent diameter and height are 30 and 10 mm, respectively. The bottom of the radiation unit is bonded onto the end face of the Langevin transducer with epoxy resin adhesive. The concave radiation face is fabricated using the grinding process, and the radius of the concave radiation face is 17.5 mm. Due to the mass effect of the radiation unit, the first-order resonant frequency of the whole ultrasonic transducer decreases to 55.1 kHz (measured using a laser Doppler vibrometer PSV-500, POLYTEC).

## 3. Experimental Phenomena and Principle

Figure 3a shows the measured surface temperature of the heating element and the vibration velocity of the focused ultrasonic radiator surface versus the input voltage of the focused ultrasonic radiator. In the experiments, the heating element was located in the focal region of the ultrasound field generated by the focused ultrasonic radiator. The surface temperature was measured when the heating source became stable, the initial temperature of the heating element was 100 °C, and the ambient temperature was 25 °C. It is seen that the focused ultrasound radiator can effectively cool the heating element. The stronger the vibration, the better the cooling effect due to the improved convective heat transfer. When the input voltage is 30Vp-p, the heating element temperature drops from 100 °C to about 55 °C. Figure 3b shows the measured surface temperature of the heating element with varying heating power versus the input power of the focused ultrasonic radiator. From Figure 3b, it is seen that with the increase in the input power of the focused ultrasonic radiator, the cooling effect increases. When the input power of the focused ultrasonic radiator is greater than 3W, the temperature drop of the heating element becomes slow due to the self-heating of the radiator and the decrease in temperature difference between the heating element and ambient temperature.

To explain the experimental phenomenon stated above, the ultrasonic field and acoustic streaming of the focused ultrasound radiator are analyzed using the finite element method (FEM) [27,28,29], and the smoke-based flow visualization method is used to observe the flow field below the focused ultrasonic radiator.

The acoustic/solid coupling module of COMSOL Multiphysics software (R6.1) is used in the calculation, and the parameters used in the computation are listed in Table 1 and Table 2. The device system is axis-symmetrical, and the physical model as a two-dimensional (2D) axis-symmetrical structure is established. The whole simulation process comprises two steps. In the first step, the acoustic field is computed in the frequency domain by applying a given vibration velocity amplitude to the actuation part. The governing equation for computing the acoustic field is
(1)$$\nabla^{2}p+{\left(\frac{\omega }{c_{0}}\right)}^{2}p=0$$
where *p* is the acoustic pressure, *ω* is the angular frequency of the acoustic field, and *c*_0_ is the sound speed. In the second step, with the simulation result of the acoustic field, the steady acoustic streaming field in the air gap is simulated on the basis of the Reynolds stress method. The governing equations for computing the acoustic streaming field are
(2)$$\nabla \cdot u_{1}=0$$
(3)$$\rho_{0}(u_{1}\cdot \nabla )u_{1}=\nabla \cdot \left\{-p_{1}I+\mu \left[\nabla u_{1}+{\left(\nabla u_{1}\right)}^{T}\right]\right\}+F_{R}$$
(4)$$F_{R}=-\rho_{0}\langle \left(u\cdot \nabla )u+u(\nabla \cdot u\right)\rangle$$
where ***u***_1_ is the acoustic streaming velocity, *p*_1_ is the pressure of air, *ρ*_0_ is the density of air, *I* is the identity matrix, *μ* is the dynamic viscosity of air, ***F_R_*** is the body force generated from the acoustic field (< > denotes the time average over a full oscillation time period), and *u* is the acoustic velocity of the acoustic field.

Figure 4a shows the computed pattern of sound intensity. It is confirmed that the sound intensity is maximum around the geometrical focal point of the concave radiation face. Figure 4b shows the computed ultrasonic streaming field in the focused ultrasonic field. It is seen that the ultrasonic streaming velocity is strong around the geometrical focal point of the concave radiation face due to the Reynolds stress gradient. The heating element can be cooled effectively when it is located in the focal region.

The smoke-based flow visualization method is used to observe the flow field below the focused acoustic radiator when the working frequency is 55.1 kHz and the driving voltage is 30Vp-p. A transparent chamber with dimensions of 500 × 500 × 600 mm was made to contain the cooling system and eliminate the influence of air flow in the room. A burning incense supplied the smoke streaming below the focused acoustic radiator. The smoke flow pattern below the focused ultrasonic radiator was observed. The photos were taken using a high-precision flow visualization system (HiSense Zyla, Dantec Dynamics A/S), and the photos with and without ultrasound are shown in Figure 5. From Figure 5a, it is seen that the smoke floats under the focused acoustic radiator smoothly when there is no ultrasound. When the focused ultrasonic radiator is turned on, the smoke flows downwards along the central axis (z-axis) of the focused ultrasonic radiator, and subsequently, vortices appear at both sides of the central axis. The observed smoke flow pattern agrees with the calculation results that the acoustic streaming eddies in the focused ultrasonic field. The asymmetry of the two vortices is caused by the heat convection in the chamber, a slight deviation of the smoke source position from the center axis, and the asymmetric smoldering.

The FEM computation and the smoke-based flow visualization experiment give the approximate position where the heating element should be placed. To find out the optimal cooling location for the heating element, the surface temperature of the heating element at different positions in the ultrasonic field was measured. In the measurement, the initial temperature was 100 °C, the ambient temperature was 25 °C, the working frequency of the focused ultrasonic radiator was 55.1 kHz, and the driving voltage was 30Vp-p. Figure 6a,b show the measured surface temperature of the heating element when the central position of the heating element is shifted along the r-axis at z = 0, and along the z-axis at r = 0, respectively. Here, the center of the spherical concave of the radiation unit is used as the original point (r = 0, z = 0). It is found that the ultrasonic cooling effect on the heating element is maximum at r = 0 and z = 0, which confirms that the cooling effect is caused by the acoustic streaming around the heating element. In Figure 6b, as the central position of the heating element is shifted along the z-axis at r = 0, the measured surface temperature change is more complex. The reason is that as the heating element moves away from the radiation surface, the acoustic energy continuously decays. Additionally, the heating element has a flat structure that can serve as a reflector and form a standing wave at specific positions, thereby enhancing the cooling effect.

## 4. Characteristics and Discussion

Fan cooling is a common method of heat dissipation. A centrifugal fan (DC brushless, BFB1012H, DELTA) is used to assess the cooling performance of the focused ultrasonic radiator in this work. As Figure 7 shows, the cooling fan is fixed with a holder; a 3D-printed jet nozzle with a diameter of 5 mm is fixed at the air outlet of the cooling fan to accelerate airflow and reduce the outlet area, and the heating element is placed right below the jet nozzle. The rotation speed of the fan is controlled through the DC power supply, and the speed of air flow is measured using an anemometer (HT-9829, Xintai Instrument Co., Ltd., Dongguan, China).

Figure 8 shows the cooling effect of the focused ultrasonic radiator and the centrifugal fan for a heating element with temperatures of 100 °C, 120 °C, and 140 °C. In the experiment, the input power of the focused ultrasonic radiator and fan is 2.6 W and 2.9 W, respectively. Figure 8a,b are the measured surface temperature of the heating element cooling using the focused ultrasound radiator and the centrifugal fan, respectively. It is seen that when the initial surface temperature of the heating element is 100 °C, the cooling effect of the focused ultrasonic radiator and the fan is similar. The surface temperature of the heating element drops from 100 °C to about 55 °C. When the initial surface temperature increases to 120 °C and 140 °C, the focused ultrasonic radiator can cool the surface temperature to 61 °C and 71 °C, respectively, while the fan can achieve 64 °C and 78 °C.

Figure 9 shows the measured cooling time of different cooling methods for the heating element once the power was turned off. The cooling time is defined as the time taken for the temperature of the heating element to drop by 90% of the total temperature drop from an initial temperature of 100 °C to a steady state temperature of 25 °C. In the experiments, the speed of air flow is 3.3 m/s and the input voltage of the focused ultrasonic radiator is 30Vp-p, which has a similar cooling effect. In Figure 9, t_1_ and t_2_ are the cooling time of the focused ultrasound radiator and fan, respectively. It is seen that the cooling time of the ultrasound is shorter than that of the fan. The cooling time of the ultrasound is about 75% that of the fan. This phenomenon is due to the different nature of the acoustic streaming fields generated by the fan and the focused ultrasound. The acoustic streaming field generated by the fan is unsteady, with a nature of intermittency [30], while that generated by the focused ultrasound is steady. A faster cooling response can reduce the duration during which the temperature rise is harmful to the working device.

The heating element is replaced with a circular heating element that has a diameter of 30 mm and the temperature is measured at various points. The distribution of temperature measurement points is shown in Figure 10. Points 1, 2, 3, and 4 are equally distributed along the radius direction, spaced 4 mm apart, and point 1 is in the center of the heating element. Figure 11a,b show the measured surface temperature at different points as the heating element is cooling using the centrifugal fan and the focused ultrasonic radiator, respectively. It is seen that when the input voltage of the focused ultrasonic radiator is 30Vp-p and the speed of air flow is 3.3 m/s, the cooling effect of the focused ultrasonic radiator is focused in the focal region, while that of air flow spreads around.

To explain the experimental phenomena for the different heat dissipation effects of the centrifugal fan and the focused ultrasonic radiator at different measurement points, the acoustic streaming fields of the centrifugal fan and the focused ultrasonic radiator were calculated using FEM. Figure 12a,b show the acoustic streaming fields of the centrifugal fan when the diameter of the heating element is 5 mm and 30 mm, respectively. It is seen that as the diameter of the heating element increases, the air flow spreads along the surface of the heating element, cooling the entire surface. Figure 12c,d show the acoustic streaming fields of the focused ultrasonic radiator when the diameter of the heating element is 5 mm and 30 mm, respectively. It is seen that as the diameter of the heating element increases, the acoustic streaming field is almost unchanged and the cooling effect is best at the center of the heating element.

## 5. Conclusions

In this work, a cooling device based on focused ultrasound for the electrical heating elements in the focal region is proposed and investigated. For a high-temperature alumina ceramic heating plate with a diameter of 5 mm in the focal region, when the input voltage is 30Vp-p and the ambient temperature is 25 °C, the focused ultrasonic radiator can drop the surface temperature of the heating plate from 100 °C to about 55 °C. The analyses of the FEM computational and flow field visualization test results indicate that focused ultrasound generates forced convective heat transfer through acoustic streaming in the focal region, which cools the heating elements effectively. When the diameter of the electrical heating element is changed from 5 mm to 30 mm, the cooling effect is similar in the focal region. Compared with a fan, the focused ultrasound radiator has a shorter cooling time and a more concentrated cooling area. Therefore, although the focused ultrasonic radiator proposed in this work has disadvantages in terms of structure and cost, it is suitable for some special environments, such as targeted heat dissipation.

## Figures and Tables

**Figure 1 micromachines-15-00116-f001:**
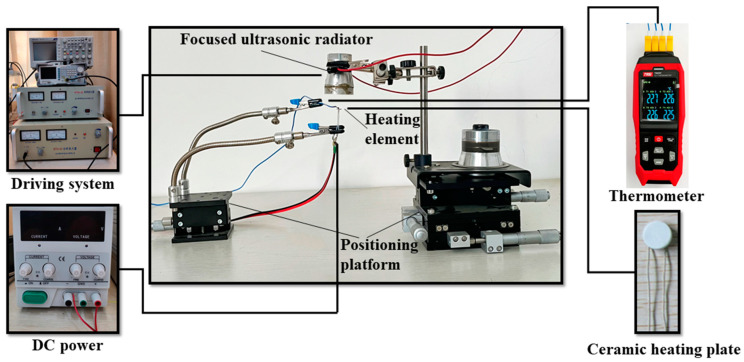
Schematic of the experimental setup to investigate the cooling effect and characteristics of the focused ultrasonic radiator.

**Figure 2 micromachines-15-00116-f002:**
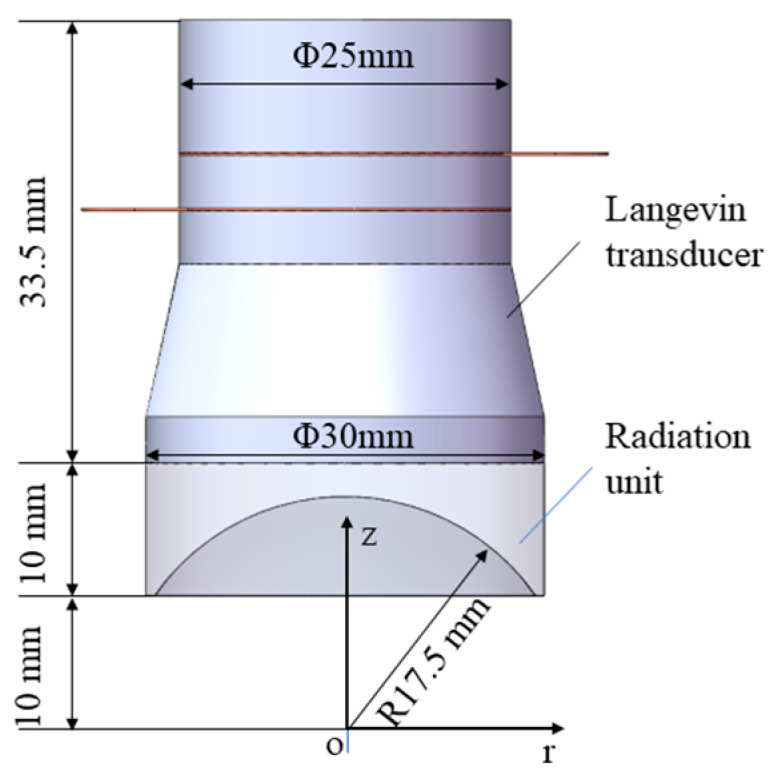
Schematic diagram of the focused ultrasonic radiator.

**Figure 3 micromachines-15-00116-f003:**
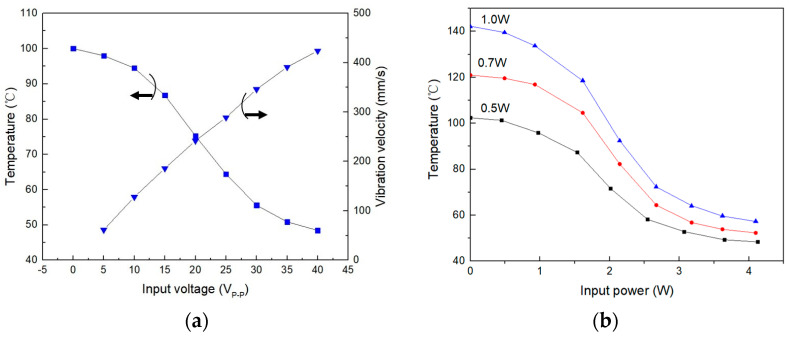
Measured cooling effects of the focused ultrasonic radiator for the heating element. (**a**) Measured surface temperature of the heating element and vibration velocity of the focused ultrasonic radiator versus the input voltage of the focused ultrasonic radiator. (**b**) Measured surface temperature of the heating element with varying heating power versus the input power of the focused ultrasonic radiator.

**Figure 4 micromachines-15-00116-f004:**
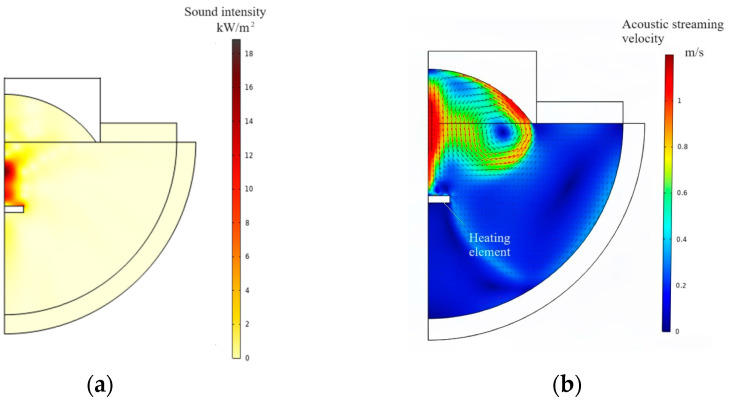
FEM computation. (**a**) Pattern of sound intensity. (**b**) Pattern of acoustic streaming field.

**Figure 5 micromachines-15-00116-f005:**
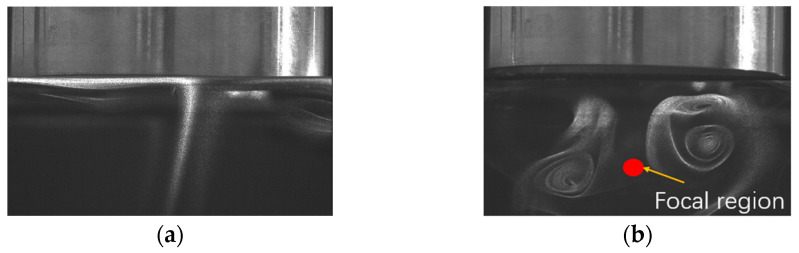
Flow field below the focused ultrasonic radiator working at 55.1 kHz. (**a**) Flow pattern without ultrasound. (**b**) Flow pattern with ultrasound.

**Figure 6 micromachines-15-00116-f006:**
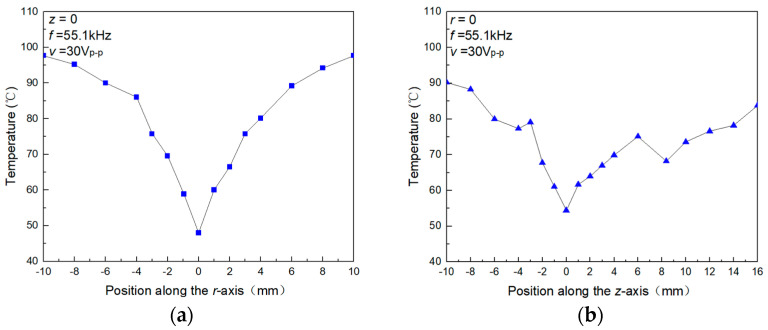
Surface temperature of the heating element at different positions in the ultrasonic field. (**a**) The central position of the sensor element is shifted along the r-axis at z = 0. (**b**) The central position of the sensor element is shifted along the z-axis at r = 0.

**Figure 7 micromachines-15-00116-f007:**
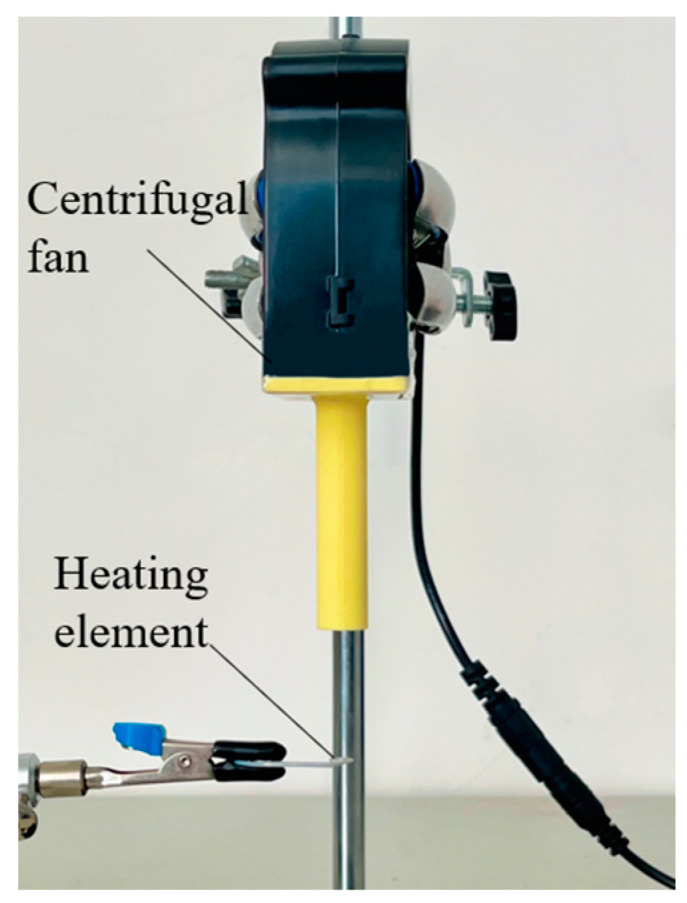
Experimental setup for fan cooling.

**Figure 8 micromachines-15-00116-f008:**
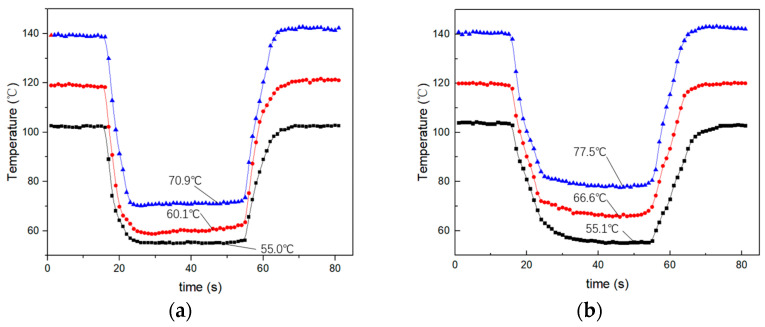
The cooling effect of the focused ultrasonic radiator and the centrifugal fan for a heating element with temperatures of 100 °C, 120 °C, and 140 °C. (**a**) The focused ultrasonic radiator. (**b**) The centrifugal fan.

**Figure 9 micromachines-15-00116-f009:**
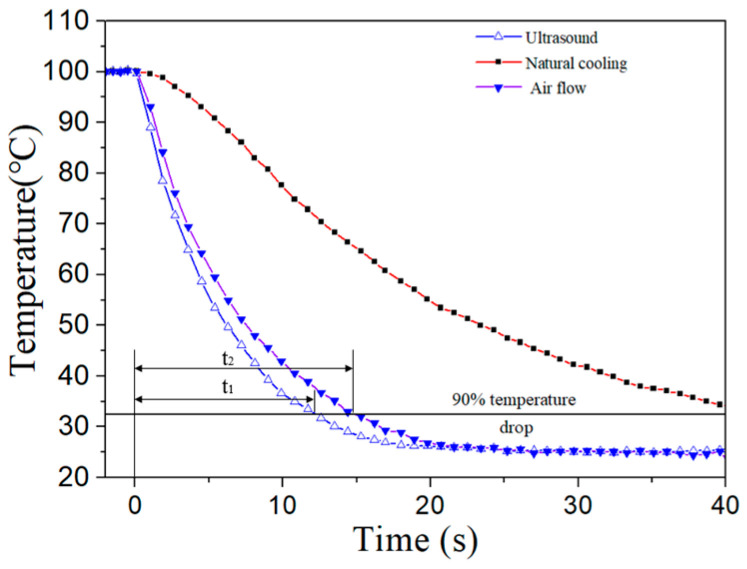
Measured cooling time of different cooling methods for the heating element after power-off.

**Figure 10 micromachines-15-00116-f010:**
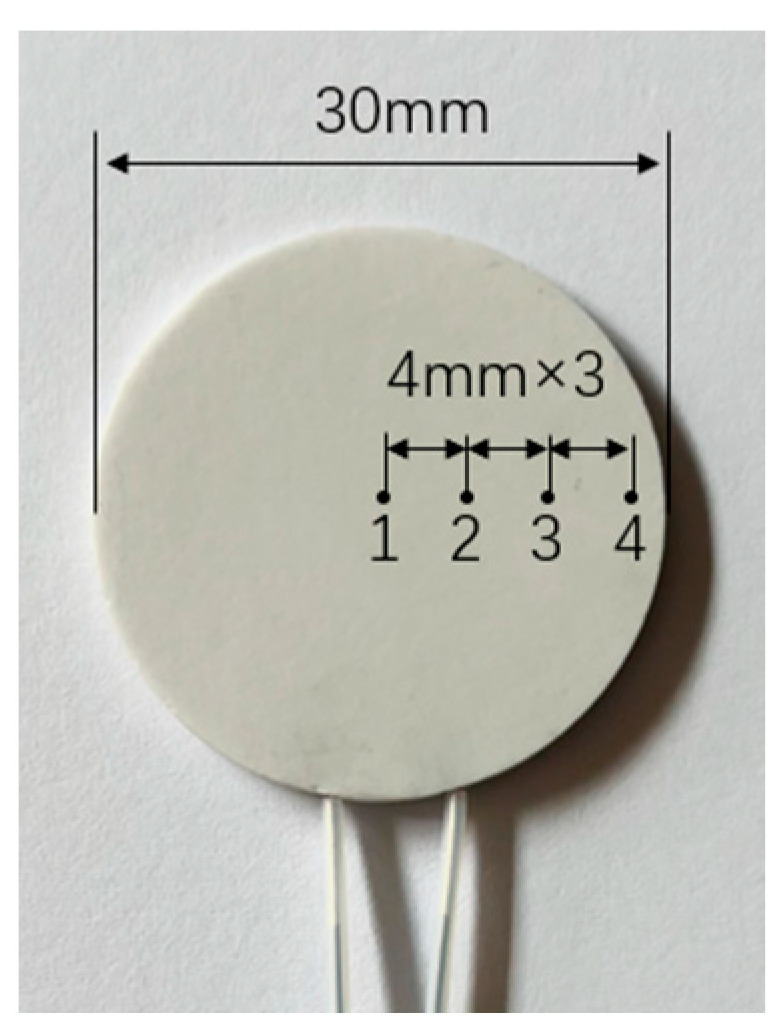
Distribution of the temperature measurement points.

**Figure 11 micromachines-15-00116-f011:**
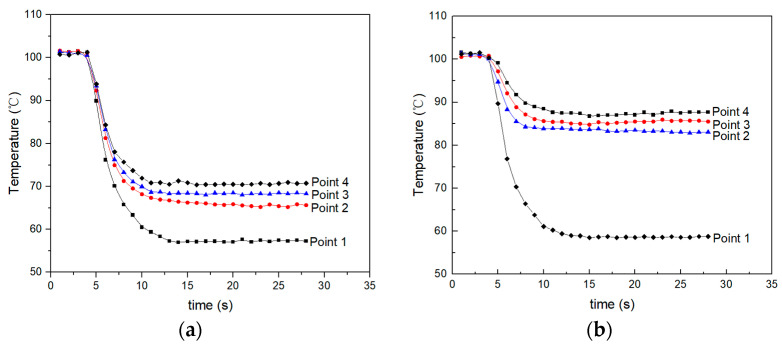
Measured surface temperature at different points as the heating element is cooling using the centrifugal fan and the focused ultrasonic radiator. (**a**) Cooling using the centrifugal fan. (**b**) Cooling using the focused ultrasonic radiator.

**Figure 12 micromachines-15-00116-f012:**
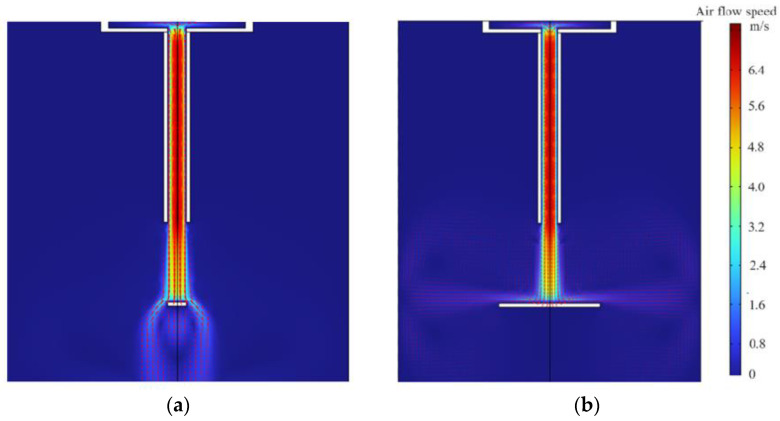
Simulated streaming fields. (**a**) Air flow streaming field when the diameter of the heating element is 5 mm. (**b**) Air flow streaming field when the diameter of the heating element is 30 mm. (**c**) Acoustic streaming field when the diameter of the heating element is 5 mm. (**d**) Acoustic streaming field when the diameter of the heating element is 30 mm.

**Table 1 micromachines-15-00116-t001:** Property constants of the acoustic medium in the ultrasonic field.

Property Constants	Value
Dynamic viscosity (kg/(m·s))	1.812 × 10^−5^
Ratio of specific heats	1.4
Heat capacity at constant pressure (J/(kg·K))	1005.6
Density (kg/m^3^)	1.204
Thermal conductivity (kg·m/(s^3^·K))	2.573 × 10^−2^
Speed of sound (m/s)	343.2
Bulk viscosity (kg /(m·s))	5.436 × 10^−6^

**Table 2 micromachines-15-00116-t002:** Dimensions and vibration excitation conditions of the ultrasonic field.

Parameters	Value
Radius of the top concave ultrasonic field *r* (mm)	14
Height of the top concave ultrasonic field *h* (mm)	7.5
Radius of the bottom hemispherical ultrasonic field r (mm)	35
Excitation frequency *f* (kHz)	55.1
Measured acceleration of the radiation face *a_z_* (km/s^2^)	0–217

## Data Availability

Data are contained within the article.

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
