# Peer review of "Study on the Cooling Performance of a Focused Ultrasonic Radiator for Electrical Heating Elements"

_micromachines, 2024, doi:10.3390/mi15010116_

Round 1

Reviewer 1 Report

Comments and Suggestions for Authors

The submitted manuscript deals with the problem of cooling hot objects employing an acoustic-field generated acoustic streaming. Even if this effect has been previously studied and it is well known, the paper is interesting and may be published, however, it is necessary to address the following issues found in the manuscript.

·       Line 44: At low sound velocity, free convection dominates, while at high sound velocity… What do the authors mean by ”sound velocity”? In acoustics, we use terms “sound speed”, and “acoustic velocity”, these terms refer to different things. The term ”sound velocity” is not used in acoustics and thus, it is ambiguous and confusing.

·       The author report that they have used COMSOL Multiphysics for the numerical simulations. However, they should specify which interfaces they used. For example, did they calculate the acoustic field employing the pressure acoustics interface of the acoustic module? Or, did they solve the compressible Navier-Stokes equations from the CFD module? And especially and most importantly, how was the acoustic streaming field calculated?

·       Figure 4(a): The figure is clearly wrong. The blue color from the colorbar refers to acoustic pressure of -400 Pa, which means, that the authors think that all the blue parts of the plot have -400 Pa of acoustic pressure, which is nonsense.  So, I would guess that the figure in fact shows the acoustic pressure amplitude (?) and the blue color refers to 0 Pa regions. This means that the colorbar is taken from another plot and one cannot believe to what is in the figure! Simply put, the figure is manipulated!

·       Figure 6(b): The dependence of the central temperature on the z-axis position does not possess one minimum, which indicates that there probably is a standing-wave component in the acoustic field – the heating element serves as a reflector of the acoustic waves and the individual minima of the temperature correspond to resonances. However, as the sound speed is temperature-dependent, the resonance conditions, and the efficiency of the system depend on the temperature. The authors should address this issue.

·       I do not understand how it is possible that the ultrasonic cooling is more effective than the one employing the fan. I suppose that the main effect causing the cooling is the forced air convection. It can be seen in Fig. 12 that the maximum streaming velocity is ca. 1m/s, and there is a recirculation zone which sucks the hot air from the heater back to the reflector and towards the radiator again. Meanwhile, the airflow speed from the nozzle is more than 6m/s and there seems not to be a recirculation zone. So, how can the ultrasonic cooling be more effective? And, would it be so if the authors moved the nozzle closer to the heater?  Or, if they decreased its radius and moved it closer? The authors should address this issue before they draw any conclusions about superiority of the ultrasonic cooling compared to the airflow one.

·      It would also be very useful, if the authors could provide the information about the power delivered to the heater, as well as the power delivered to the ultrasonic cooling system.

Comments on the Quality of English Language

The manuscript should be checked by a native English speaker. A few examples: Line 171 – fan is fix, Line 172 – was fix, Line 291 - is short than, Line 229 – acoustic flow field of the centrifugal fan, Line 232 – which cooling the, Line 257 – has and shorter, etc.

Author Response

Dear editor and reviewers,
  Thanks for your review and valuable comments, which are quite helpful for us to improve our manuscript. According to your comments, we have revised our manuscript (detailed modifications are highlighted in the revised manuscript).
The attachment is a detailed response.

Reviewer 2 Report

Comments and Suggestions for Authors

The author developed a focused ultrasonic radiator for cooling small heating element. Temperature changes were measured, and acoustic streaming patterns were simulated. This work is quite interesting. However, some parts needs revision before publication.

For the fair comparison of cooling effect, the electrical output of focused ultrasonic radiator and centrifugal fan. In addition, the nozzle design could also be optimized for better cooling.

Although this investigation proved that focused ultrasound cooling is effective, the final conclusion of the focused ultrasonic radiator proposed in this work is well suitable for cooling the small heating elements may not be very strong. Considering the high expense of ultrasound radiator and its driving equipment as well as the large size, the proposed approach may not be used widely in practice. In some special environment, it may be a good choice.

Limitation of ultrasonic cooling should also be discussed in the manuscript.

Line 13 change at the focal region to in the focal region

Line 107 change leaded to to led to

Line 172 change fix to fixed

Line 169-175 why the authors think the air flow is focused?

Comments on the Quality of English Language

minor changes according to my specific comments

Author Response

(The authors gave the same response as above.)

Round 2

Reviewer 1 Report

Comments and Suggestions for Authors

In my previous letter, I have pointed out that the authors use term “sound velocity,” which is not used in acoustics and is misleading. In acoustics, we use the terms “sound speed,” which means the wave speed, and “acoustic velocity,” which refers to the oscillation velocity of the particles forming the fluid.  On lines 44 and 45 the authors changed the terms “sound velocity” to “sound speed,” however, they should have changed it to “acoustic velocity,” because they can influence the acoustic velocity by increasing the input power of the transducer, whereas the sound speed can only be changed by changing the fluid temperature.

Comments on the Quality of English Language

English used in the manuscript is quite good.

Author Response

Dear editor and reviewers,
Thanks for your review and valuable comments, which are quite helpful for us to improve our manuscript. According to your comments, we have revised our manuscript (detailed modifications are highlighted in the revised manuscript).
Our response to your comments is appended below.

  1. In my previous letter, I have pointed out that the authors use term “sound velocity,” which is not used in acoustics and is misleading. In acoustics, we use the terms “sound speed,” which means the wave speed, and “acoustic velocity,” which refers to the oscillation velocity of the particles forming the fluid. On lines 44 and 45 the authors changed the terms “sound velocity” to “sound speed,” however, they should have changed it to “acoustic velocity,” because they can influence the acoustic velocity by increasing the input power of the transducer, whereas the sound speed can only be changed by changing the fluid temperature.

ANS: Thanks for your comments. This part of the content has been replaced according to the suggestion of the other reviewer,and we will pay attention to the difference between “acoustic velocity” and “sound speed” in future writing.

Reviewer 2 Report

Comments and Suggestions for Authors

Line 42-46 this explanation is wrong

Line 47-52 the phenomenon does not occur only when the vibration frequency was 30kHz. which is type of vibration source, planar or focused one?

Figure 3 specify when the temperature at the heating source becomes stable. Figure 4a the distribution of acoustic intensity is wrong

Figure 5 why the flow pattern with ultrasound is not symmetric? Please explain

Line 184-188 many grammar errors, and the authors dont give reasonable explanation

Comments on the Quality of English Language

need polish

Author Response

Dear editor and reviewers,
Thanks for your review and valuable comments, which are quite helpful for us to improve our manuscript. According to your comments, we have revised our manuscript (detailed modifications are highlighted in the revised manuscript).
Our response to your comments please see the attachment.
